# Resilience of the Asian atmospheric circulation shown by Paleogene dust provenance

A. Licht[1,2,3], G. Dupont-Nivet[2,4,5], A. Pullen[6,7], P. Kapp[6], H.A. Abels[8], Z. Lai[9], Z. Guo[5], J. Abell[6] & D. Giesler[6]

The onset of modern central Asian atmospheric circulation is traditionally linked to the interplay of surface uplift of the Mongolian and Tibetan-Himalayan orogens, retreat of the Paratethys sea from central Asia and Cenozoic global cooling. Although the role of these players has not yet been unravelled, the vast dust deposits of central China support the presence of arid conditions and modern atmospheric pathways for the last 25 million years (Myr). Here, we present provenance data from older (42–33 Myr) dust deposits, at a time when the Tibetan Plateau was less developed, the Paratethys sea still present in central Asia and atmospheric $pCO_2$ much higher. Our results show that dust sources and near-surface atmospheric circulation have changed little since at least 42 Myr. Our findings indicate that the locus of central Asian high pressures and concurrent aridity is a resilient feature only modulated by mountain building, global cooling and sea retreat.

[1] Biodiversity Institute, University of Kansas, Lawrence, Kansas 66045, USA. [2] Institute of Earth and Environmental Sciences, Universität Potsdam, Potsdam 14476, Germany. [3] Department of Earth and Space Sciences, University of Washington, Seattle, Washington 98195, USA. [4] Géosciences Rennes, UMR CNRS 6118, Université de Rennes, Rennes 74205, France. [5] Key Laboratory of Orogenic Belts and Crustal Evolution, Peking University, Beijing 100871, China. [6] Department of Geosciences, University of Arizona, Tucson, Arizona 85721, USA. [7] Department of Earth and Environmental Sciences, University of Rochester, Rochester, New York 14627, USA. [8] Department of Geosciences and Engineering, Delft University of Technology, Delft 2628 CN, The Netherlands. [9] School of Earth Sciences, China University of Geosciences, Wuhan 430074, China. Correspondence and requests for materials should be addressed to A.L. (email: licht@uw.edu).

The inner-Asian high-pressure system controls aridification at the continental scale, and its dynamics have profoundly influenced Asian ecosystems[1]. Its current mean position between 25°–45° N latitude is generated by the descending branch of the tropical atmospheric circulation cell (also known as the Hadley cell), and is migrating poleward under global-warming stress[2]. Two major atmospheric configurations have been posited for Quaternary time; (a) during interglacials, the subtropical jet stream remains south of the Tibetan Plateau in winter and the high-pressure system is located over Mongolia (Fig. 1a). Dust-storms originate from the break-up of the high pressures during spring and blow southeastward across central China[3,4]. During summer, the jet stream is located north of the Tibetan Plateau, allowing monsoonal moisture penetration inland[5]; (b) during glacial periods, inner Asia is characterized by more persistent and higher pressures and the jet stream is largely restricted to the south of the Tibetan Plateau[4,6–9]; together with weaker insolation forcing, this results in reduced inland penetration of monsoonal moisture[5] (Fig. 1b). Enhanced cooling over the North Atlantic shifts pressure maxima to northwest Eurasia and results in a stronger penetration of surface westerlies in central Asia, which transport dust from the northern margin of the Tibetan Plateau[6–10].

Dust accumulation initiated on the Chinese Loess Plateau (CLP) 25–22 million years (Myr) ago and accelerated ∼3 Myr ago. These observations have been used to argue for the early Miocene initiation, and subsequent strengthening of the Asian high-pressure system and concurrent aridity[1,11]. Development of the Asian high-pressure system has been attributed to three processes. First, late Cenozoic global cooling and onset of glacial conditions favoured high pressures at mid latitudes; the Quaternary development of the Eurasian ice-sheet, in particular, modified the North Atlantic meridional circulation and enhanced surface westerlies in central Asia[6,7]. Second, the growth of the Mongolian and Pamir-Tibetan plateaus may have enhanced

westerly winds and anchored the position of the subtropical high over Mongolia during interglacials[12]. Finally, the progressive (50–20 Myr ago) retreat of the Parathethys sea from central Asia would have increased seasonal thermal contrasts and favoured winter high-pressure systems north of Tibet[13,14].

Empirical constraints on pre-Miocene atmospheric circulation are scarce, although fundamental to assessing the extent to which orographic growth, retreat of the Parathethys and late Cenozoic global cooling impacted atmospheric circulation in Asia. One hypothesis is that before Miocene time, the presence of the Parathethys sea in western China[15] and less extensive and/or lower elevation Tibetan and Mongolian plateaus[12,16] would have resulted in a weak central Asian high-pressure system and favoured penetration of moisture into western and central Asia. High pressures would have been shifted to eastern China, weakening East Asian monsoons and favouring a southeastward shift of regional aridity[13,14] (Fig. 1c); aridification of central Asia would have been later enhanced after both the retreat of the Parathethys sea and the uplift of the Pamir-Tibetan Plateau in the late Oligocene–early Miocene[17]. A second hypothesis for pre-Miocene time is that the synoptic atmospheric system was shifted by 20° of latitude to the north and that the intertropical convergence zone (ITCZ) was anchored to the Tibetan Plateau[18], favouring wetter conditions and penetration of trade winds into central Asia (Fig. 1d). Late Cenozoic cooling and expansion of northern hemisphere ice-sheets would then have shifted the zone of peak sea-surface temperatures southward and led to the drift of the ITCZ to its modern location[18]. The two hypotheses make fundamentally different predictions about wind patterns as well as the geographical position and timing of initial aridification in the Asian interior (Fig. 1c,d).

The presence of Eocene dust deposits in the Xining Basin, at the southwestern margin of the CLP (Fig. 2), suggests that desert expansion and dust generation in central-eastern Asia may have initiated much earlier than previously thought[19]. Constraining

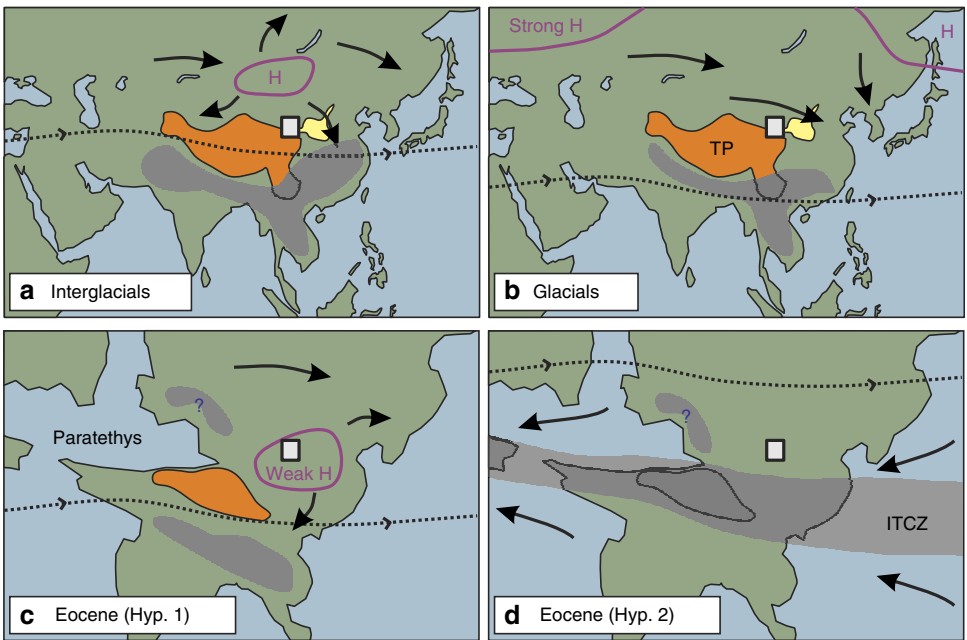

**Figure 1 | Winter atmospheric circulation in Asia during Quaternary and Eocene times.** Graph shows the location of pressure highs (H, in purple) and associated winter winds (black arrows), the average annual position of the subtropical jet stream (dashed line) as well as the areas of significant summer rainfall (blue shaded areas) for interglacials (**a**), glacials (**b**) and hypothesized circulations for the Eocene (**c,d**). Xining area (square) lays at the interface between the Chinese Loess Plateau (in yellow) and the Tibetan Plateau (TP, in orange). Displayed on a middle Eocene paleogeographic reconstruction[15], Eocene hypothesis 1 (**c**) proposes a semi-permanent subtropical high in eastern China[13,14]; Eocene hypothesis 2 (**d**) proposes a 20° northward shift of the synoptic atmospheric system and the anchoring of the ITCZ to the Tibetan Plateau[18].

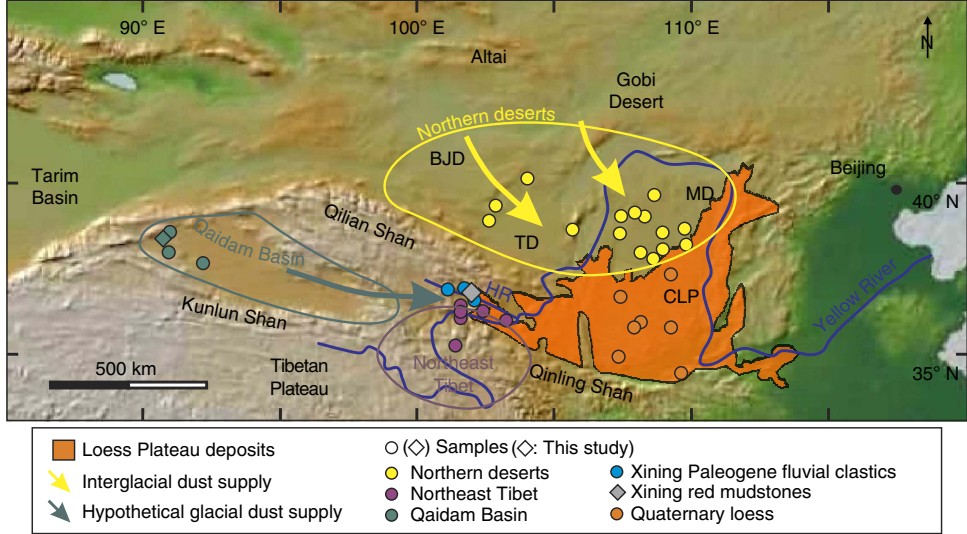

**Figure 2 | Dust supply patterns to the Chinese Loess Plateau.** The figure shows the modern, simplified dust-storm tracks (yellow arrows), throughout the Mu Us (MD), Tengger (TD) and Badan Jaran (BDJ) deserts, the hypothetical 'glacial' dust-storm track (green arrows), throughout the Qaidam Basin[10], the provenance regions for the Eocene dust (coloured ellipses) and the samples used in this study (description, location and references are provided in Supplementary Data 1). HR, Huangshui River, tributary of the Yellow River (both shown by blue lines) incising the Xining Basin.

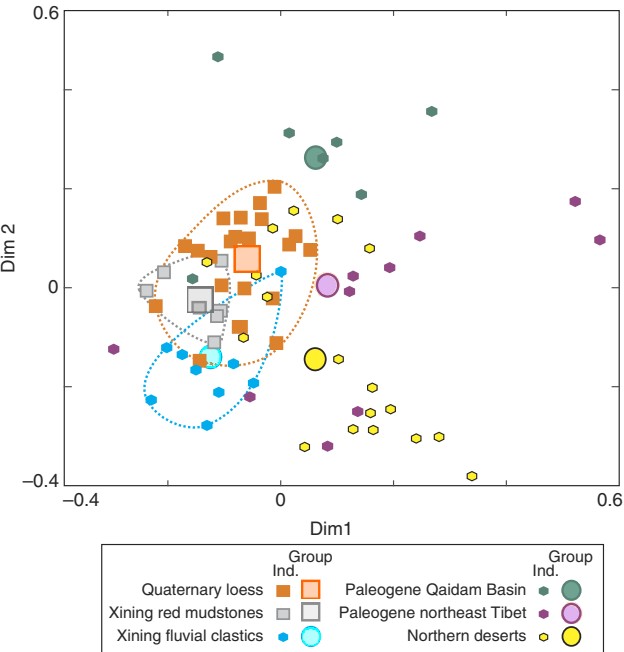

**Figure 3 | Dissimilarities between individual samples and age compilations.** Shown on a multidimensional scaling map[36], a visual way to assess the misfit between age distributions using the KS statistic as the dissimilarity measure. Axes are in dimensionless 'KeS units' (0 < KS < 1) of dissimilarity between samples. Final 'stress' value is 0.13, indicating a fit between 'good' and 'poor'[36]. Ranges of variation for individual samples of Quaternary loesses, Paleogene red mudstones and fluvial clastics of the Xining Basin are highlighted by dashed lines. Note that red mudstones individual samples fall within Quaternary loesses on the map, indicating low dissimilarity, but are distinct from Paleogene fluvial clastics of the Xining Basin. Group, age compilations; Ind., individual samples.

the provenance of these strata provides the opportunity to reconstruct surface-wind pathways, and thereby assess both pre-Miocene hypotheses. The Xining Basin is part of a large Paleogene-Miocene basin system; it includes characteristic red

mudstones intercalated with layers of gypsum that can be traced for hundreds of kilometres and have been dated between 42–33 Myr (refs 20–22). Sedimentological studies have shown that the gypsiferous layers formed during periods of relatively high water supply in shallow saline lakes, whereas the mudstones were deposited in arid playa environments; these wet–dry alternations followed obliquity cycles resolved by magnetostratigraphy[20,21]. Rare, isolated alluvial packages with channel bodies and sand-flat deposits are interbedded within the red mudstone–gypsum successions and indicate occasional fluvial transport[23]. U-Pb ages of zircons in the sandstone intervals indicate a proximal origin in the surrounding highlands, the Qilian and western Qinling mountain ranges[23]. However, evidence of aeolian abrasion on quartz grains and grain-size distributions—similar to those of modern loess—indicate that the red mudstone playa strata received aeolian detritus[19].

Zircon U-Pb age distributions of detrital zircons from loess samples are useful tracers of dust provenance, from which past surface-wind directions can be inferred[4,10,24–26]. Modern laser-ablation techniques allow the dating of aeolian zircons that are commonly > 12 μm; zircon age distributions are therefore representative of the coarse dust fraction that is transported by wind in low-level suspension clouds[4,26]. Zircon populations from Asian dust sources display the same age peaks (225–330 Myr, 380–500 Myr, 750–1,100 Myr and 1,500–2,500 Myr), yet in different proportions[8,24–26]; identifying aeolian zircon sources therefore requires a high number (n > 800) of zircon ages per region to ensure that source age distributions are statistically representative and are not artefacts of sub-sampling effects[26] (see the 'Methods' section).

With the aim of reconstructing the direction(s) of pre-Miocene winds blowing dust into the Xining Basin, we determine U-Pb ages on a total of 1,640 zircons from 8 Paleogene samples. Our new zircon age distributions show that lithogenic dust sources and near-surface atmospheric circulation have changed little since 42 Myr and argue for the long-term stability of the central Asian synoptic-level atmospheric circulation.

## Results

**U-Pb age distributions.** Zircon age distributions from red mudstones are statistically different from those of Paleogene

fluvial sandstones from the same basin[23] (Fig. 3) based on the Kolmogorov–Smirnov (KS) statistic, a common dissimilarity measure for age distributions[26]. A single provenance for the mudstones and for the Paleogene fluvial sandstones is unlikely because: (1) the 1,500–2,500 Myr zircon population is almost two times more abundant in the combined age distribution of fluvial sandstones than in the one of mudstones; and (2) the mudstone combined age distribution displays a well-defined age peak at 800–1,100 Myr, twice the size that is observed for the one of fluvial sandstones (Fig. 4). These observations indicate an additional, non-fluvial supply from a different source with a different age distribution than the local supply from the proximal highlands and corroborate the aeolian nature of the red mudstones[19].

Surrounding the Xining Basin, three potential—and more distal—sources for the Paleogene mudstones can be distinguished as follows (Fig. 2): (1) the northern deserts, corresponding to the low-relief areas extending north of the modern CLP that include the sandy Badan Jaran, Tengger and Mu Us Deserts; (2) the Qaidam Basin, extending between the Kunlun and Qilian mountains, located west of the Xining Basin; and (3) Northeast Tibet, and the Songpan-Ganzi terrane region. These three source areas contributed to the Quaternary supply of the CLP, with Northeast Tibet being prominent due to the role of the Yellow River, which transports sediment to the north that is later reworked by wind[24–26]. A contribution from the Tarim area, as proposed by others for the Pliocene loess[27], seems unlikely due to the episodic presence of a shallow epicontinental sea in that area for part of the interval of study[15]; moreover, we argue that any coarse (>12 μm) aeolian zircon supply to the Xining Basin from the Tarim area would have to transit in low-level suspension clouds through the Qaidam Basin and therefore its signature should be represented in the zircons of this region (Fig. 2).

To identify the main Paleogene dust contributor, we compared mudstone age distributions with age compilations from these three areas (Supplementary Table 1). The aeolian source(s) must include a smaller proportion of 1,500–2,500 Myr ages and a

higher proportion of 800–1,100 Myr ages to achieve the age distributions of the red mudstones, combined with Paleogene fluvial input. In this comparison, we favour data from new and published Paleogene samples that are not affected by changes in provenance signature as a result of Neogene surface uplift and denudation in northern Tibet[23,28,29]. The Qaidam Basin age compilation exhibits few ages >1,500 Myr (<10%) and a significant proportion of ages distributed around 900 Myr (Fig. 4), thus fitting the requirements for the additional source. The proportion of 1,500–2,500 Myr ages is too high in the northern deserts (>50%) and northeast Tibet (>35%) age compilations; 800–1,100 Myr ages are scarce in both of these regions. These observations suggest that northeast Tibet and the northern deserts are unlikely contributors and that the Qaidam Basin is the best source to explain the mudstone age distribution when mixed with local fluvial input.

We applied an iterative mixture modelling strategy[26] to determine the combination of the four potential source regions (local highlands, northeast Tibet, northern deserts and Qaidam Basin) that best fits the combined age distribution of Paleogene mudstones. The two measures of the best fit display similar values (see 'Methods'; Supplementary Table 2). Age distributions for these combinations are statistically similar to the one of red mudstones in the sense of the KS statistic, and uncertainty around these combinations is very low (see the 'Methods' section). Both indicate a best fit for a contribution from local highlands of 70–75%, a contribution from the Qaidam Basin between 20–25%, and the two other provinces contributing the remainder (4–8%). Mudstone provenance is thus best explained by recycling of local fluvial input and longer-distance aeolian supply from the Qaidam Basin.

## Discussion

Our results confirm the aeolian nature of the red mudstones and show that climatic conditions were arid enough to allow wind deflation and dust transport in central Asia since at least 42 Myr. They corroborate studies suggesting aeolian dust sedimentation on the CLP back to the early Cenozoic[30] as well as desert expansion in central Tibet[31]. The sedimentary budget for the red mudstones is similar to the budget of the Quaternary CLP, where the Qaidam Basin contributed ∼20% of the supply and most of the loess (60–70%) was recycled from fluvial deposits—albeit via different fluvial systems[25,26]. Our results thus indicate that the main dust supply mechanisms in central China have changed little over the past 42 Myr. Our provenance analysis indicates the persistence of surface westerly, dust-generating winds blowing through the Qaidam Basin region and along the northern margin of the Tibetan Plateau[9,10]. Along with paleo-wind reconstructions of Upper Cretaceous[32], Mio-Pliocene[27] and Quaternary[7,9,10] aeolian deposits, our results emphasize the long-term prominence of surface westerly winds in central Asia and corroborate the persistence of high-pressure belts at these latitudes. These reconstructions do not support any of the previously proposed scenarios for Paleogene atmospheric circulation because moisture penetration into western and central Asia would reduce aridity and dust production, and transport along the northern margin of Tibet, and a subtropical high restricted to eastern China would dampen westerly circulation in the study area (hypothesis 1); similarly, if the ITCZ was anchored to the Tibetan Plateau, this would dampen dust production in west and central Asia and favour easterly rather than westerly surface winds (hypothesis 2).

Rather, our results underscore the resilience through time of the descending branch of the Hadley cell in central Asia, around 25°–45° N paleolatitude. Marked aridity in central Asia, despite

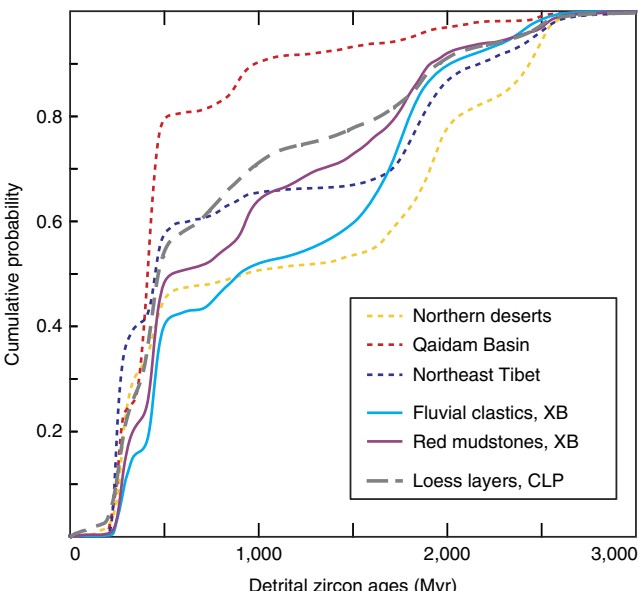

**Figure 4 | Age distributions of Paleogene Xining sediment compared with modern loess and Paleogene dust sources.** Cumulative probability density plots for Paleogene red mudstones and fluvial sandstones from the Xining Basin, Quaternary loess of the CLP, Qaidam Basin, Northeast Tibet and northern deserts. XB, Xining Basin; CLP, Chinese Loess Plateau.

evaporation in the Eocene Parathys, particularly during the summer[33], requires subtropical high pressures throughout most of the year in western China to dampen local rainfall[14]. These high pressures could explain westerly dominated surface circulation along the northern margin of Tibetan Plateau and indicate that the shallow (commonly <100 m deep in central Asia[15]) Parathys Sea did not provide a sufficient thermal reservoir to impact inner-Asian atmospheric circulation.

The similarity between Eocene and Quaternary atmospheric circulation is further emphasized by the Eocene ephemeral lacustrine expansions in the Xining Basin that resulted in orbitally controlled gypsum deposition[20,21]. They either suggest temperature-enhanced moisture supply along the existing westerly track and/or weakening or shifting of the subtropical high in west China, allowing more moisture penetration from the Pacific (and thus reflecting enhanced monsoonal penetration) or from the westerlies[15,19,34]. Regardless of exact location of the moisture source, these orbitally controlled wetter periods mirror central Asian interglacials, with enhanced moisture supply and weakened dust input from the westerlies[5-10].

Our findings thus indicate Quaternary-like atmospheric dynamics for the late Eocene in central Asia, with the prominence of arid conditions and surface westerly winds orbitally intermingled with periods of enhanced moisture supply and decreased dust production along the westerly track. They show that the marked changes in palaeogeography, topography and global climate during the Cenozoic had little effects on the synoptic-level atmospheric circulation in central Asia. This early Cenozoic origin for the central Asian high-pressure system does not question the numerous paleoclimatic proxy records showing aridification and desert expansion throughout central Asia since then[1,11,17,20]. It does, however, suggest that this long-term aridification trend is linked to changes in the hydrological cycle and in moisture availability at the surface and atmosphere rather than major re-organization of wind patterns and moisture transport pathways. Changes in $pCO_2$, the retreat of the Parathys sea, and recent (Miocene–present) surface uplift of in northern Tibet might have modulated moisture transport along existing Paleogene atmospheric patterns by limiting the amount of available moisture along both surface westerlies and monsoonal tracks[15,19,34]. However, our results show that these radical changes had little impact on the synoptic atmospheric circulation itself.

In addition, our results have implications for Quaternary loess provenance interpretations. Despite unchanged dust suppliers, Quaternary loess displays zircon age distributions that slightly differ from Paleogene Xining red mudstones, with a commonly higher proportion of <500 Myr ages, intermediate between red mudstones and Qaidam Basin distributions (Figs 3 and 4). This difference can be explained by two processes. (1) Fluvial input into the Quaternary Loess Plateau is dominated by supply from the Yellow River, the drainage of which extends much farther south in northeast Tibet than the proximal highlands surrounding the Xining Basin[25]. Northeast Tibet-derived sandstones (Fig. 4) and the Quaternary Yellow River sedimentary load[25,26] display a young (<500 Myr) zircon age population that is 25% more prominent than in Xining Paleogene sandstones. An increased contribution of northeast Tibetan zircons by fluvial recycling since the set-up of the Yellow River in the Pliocene[25] could thus explain the observed distribution differences. (2) Miocene exhumation of northeast Tibet led to the tectonic segmentation of the Xining Basin[28], where red mudstone units are today deeply incised by the Huangshui River, a tributary of the Yellow River[35]. Erosional recycling of these older dust deposits in the Yellow River drainage, mixed with aeolian input from the Qaidam Basin, could in addition explain why Quaternary loess age populations display intermediate

distributions between red mudstones and Qaidam Basin distributions. In that sense, both potential processes show that Tibetan uplift impacted loess provenance by modifying the sources of the fluvial sediment later recycled by winds, yet without altering the Asian atmospheric circulation.

## Methods

**U-Pb dating of aeolian zircons.** U-Pb ages of $n = 100–300$ detrital zircons (grain-size: 10–45 µm) were determined from seven red mudstone layers ranging in age from 42–33 Myr (refs 21,22). We also determined U-Pb zircon ages from one Oligocene sandstone of the upper Ganchaigou Formation in the Qaidam Basin to increase the size of the U-Pb age compilation of this area for comparison[29]. Zircon crystals were extracted by traditional methods of heavy mineral separation at the Arizona LaserChron center and U-Pb ages were generated using laser-ablation multi-collector inductively coupled plasma mass-spectrometry, with a Nu Plasma ICP-MS coupled to a Photon Machines Analyte G2 193 nm Excimer laser[10,26]. The coarse-grained sample (Oligocene sandstone) was ablated using a 30 µm laser beam diameter at 7.0 mJ (constant energy) with 94% laser energy at 7 Hz. Finer-grained samples (that is, red mudstone samples) were analysed using a 12 µm with same laser fluence. Sample descriptions with age, GPS coordinates and zircon U-Pb ages are available in Supplementary Data. Detailed sedimentological descriptions of the sampled sections can be found in previous articles[21,22,29].

**Statistical treatment.** The age distributions of the individual samples are shown in Supplementary Fig. 1 of the Supplementary Information. All display the four main zircon populations that are commonly found in central Asian age distributions of detrital and igneous rocks (225–330 Myr, 380–500 Myr, 750–1,100 Myr and 1,500–2,500 Myr), but differ in the relative contribution of these age components[4,10,24,26]. Interpreting the contribution of these components is a necessary step to highlight provenance differences in central Asia, where similar age peaks are shared by all dust sources and requires a high number ($n > 800$) of zircon ages per region to ensure that the relative age distributions from the sample aliquots are not determined by sub-sampling effects[26]. Thus, Paleogene mudstone U-Pb ages were grouped into one single age distribution to ensure a large number ($n = 1,246$). This age distribution was first compared with U-Pb ages from Paleogene fluvial sandstones of the Xining Basin[23] ($n = 830$) and a compilation of published U-Pb ages from Quaternary loess of the CLP ($n = 2,285$). It was then compared with age compilations of the three potential, more distal sources: Qaidam Basin, northeast Tibet and the northern deserts (Fig. 2). To take into account changing topography and denudation in Tibet and the related changes in provenance signature in the source areas, we compiled U-Pb ages from sandstones in the Qaidam Basin ($n = 892$) and in the basins at the northeastern margin of Tibet ($n = 868$, excluding the Xining Basin) of Eocene to earliest Miocene age only. Unfortunately, Paleogene clastic rocks are absent in the northern deserts, but modern sands are mainly derived from recycling of local, pre-Cenozoic substratum as well as from minor recycling of sediment brought by Tibetan-sourced and Altai-sourced rivers[4]. We thus used a compilation of U-Pb ages from modern and pre-Cenozoic deposits from the northern deserts ($n = 1,753$) as an attempt to reflect their provenance signature in the Paleogene; using alternative pre-Cenozoic only compilations does not change the forthcoming observations (not shown). Age distributions for the all the compilations are given in Supplementary Fig. 2 and data sources are provided in Supplementary Table 1.

The iterative mixture modelling strategy of Licht et al.[26] is designed to statistically address the contribution of four different sources with known age distributions to a fifth known distribution (here, for the Paleogene mudstones). For each possible combination of the four source regions, we modelled N synthetic age distributions of 800 ages by randomly picking the necessary amount of ages in each source data set ($N = 200$, that is large enough to reproduce the potential variability in zircon sub-sampling during transport and consecutive mixing). We then calculated the dissimilarity between each of these N synthetic age distributions and the Paleogene mudstones age distribution, using the KS statistic as the dissimilarity measure (Supplementary Fig. 3). There are two ways to quantify the combination that best fits to the red mudstones (Supplementary Table 2). First, the best combination can be seen as the one for which the minimum dissimilarity $\delta_{min}$ value is reached among all the synthetic possible combinations and their N replicates. Alternatively, the best combination can be seen as the one for which the average dissimilarity $\Delta_{min}$ value among the N synthetic age distributions per potential source combination is the lowest. The latter approach takes into account the reproducibility of the dissimilarity values per source combination but minimizes the potential effect of sub-sampling bias in the contribution of the source provinces[26]. An approach to determine the uncertainty around these best fit values is to look at the range of combinations for which >10% (or 50, or 75%, depending on the required precision) of the N random synthetic distributions are statistically similar to the loess (here, in the sense of the KS statistic at the 95% confidence level), highlighted in Supplementary Fig. 3.

**Data availability.** The authors declare that all data supporting the findings of this study are available within the Supplementary Information files.

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

## Acknowledgements

A.L. was funded by the University of Kansas and by the E.U. horizon-2020 project 656731 'ECCAMETT'. G.D.-N. acknowledges support from the A. von Humboldt foundation, FP7 CIG grant 294282 'HIRESDAT' and Horizon 2020 ERC grant 649081 'MAGIC'. This work has also been supported by the University of Arizona and by the NSF AGS-1203427 and AGS-1203973 grants. We thank N. Mclean, A. Möller, Y. Donnadieu, M. Pecha, G. Simpson, I. Nurmaya and C. White for prolific discussions and assistance in the lab.

## Author contributions

A.L. and G.D.-N. conceived and designed the study; A.L., P.K., A.P., J.A. and D.G. performed the experiments; H.A.A., Z.L. and Z.G. contributed materials/analysis tools; all authors analysed the data; A.L. wrote the paper with the help of the co-authors.

## Additional information

**Competing financial interests:** The authors declare no competing financial interests.

**How to cite this article**: Licht, A. *et al.* Resilience of the Asian atmospheric circulation shown by Paleogene dust provenance. *Nat. Commun.* 7:12390 doi: 10.1038/ncomms12390 (2016).

