## [Peer Review File · Nature Communications]

Reviewers' comments:

Reviewer #1 (Remarks to the Author):

This is an exciting paper on one of the central questions in Cenozoic climate and landscape evolution: aridity onset and set up of current circulation over Asia. The manuscript presents new single-grain provenance data from Eocene-Oligocene deposits recently identified as some of the earliest aeolian sediments in central-eastern Asia. These deposits have not universally been accepted as aeolian, despite some clear evidence presented previously, so the addition of provenance data is timely. The provenance data also serve to address two competing scenarios for Asian climatic evolution, which has implications for understanding the impact of tectonic and sea level drivers of climatic evolution. The subject matter is therefore ideal for publication in Nature Communications as it appeals to a wide range of Earth Scientists and climate researchers and underpins the climatic evolution of an important and highly populated region. If anything the authors undersell this in the submission. For example, they do not clearly state just how fundamentally different and incompatible the Eocene-Oligocene climate scenarios outlined in the paper are. They could also make it clearer from the outset of the submission that it is not just circulation but the timing of initial aridification of the Asian landmass that is in question here. I have some comments, as outlined below, and most of these are requests for clarifications or further explanations, although I also suggest some simple further modelling. Generally, the manuscript suffers a bit from being a little too 'cut down'. There is a lack of detail at key points and some of the interpretations appear to be less well substantiated in the text, even if they are once you dig a little deeper. These need to be explained more thoroughly and clearly in the text. I realise there is a strict page limit here but to me a lot more of the Methods section in the manuscript could be incorporated into the supplementary file to save manuscript space. The Methods section actually within the paper should be a very short summary. At the moment it's taking up more space than needed at the expense of more clear and reasoned analysis of the data. Also, the Discussion is rather disjointed at present. There is a lack of clear flow or narrative and the paragraphs not linked well together. Some conclusions seem less well supported while others are well supported but not always clearly explained. Overall, the Discussion presents lots of interesting ideas but I am not 100% sure what is specifically new in this submission, and what has not been already proposed in Licht et al., 2014 Nature 513, 501-506. The dataset is clearly new (provenance data from the Palaeogene loess deposits), but the proposal that this is loess and that it supports monsoonal and westerly flow (e.g., current/Quaternary synoptic conditions) much earlier than previously proposed were already in Licht et al., 2014. I think the dataset here is extremely important, and has big implications, however, I would like the authors to really make clear what is actually newly proposed from this data, rather than corroborating their previously proposed hypothesis. Overall I would recommend that if the authors can clarify the specific new findings here, and clarify/improve some of the writing while address some of the questions over the analysis, then this would be an exciting manuscript, and certainly publishable in Nature Communications.

At one or two points the choice of references does not seem completely appropriate. The authors need to take care that the right references are used to support their statements. For example: Reference 4 - this reference seems out of place in the text (line 49) as it's not directly related to summer monsoon front movement. Was this reference supposed to go elsewhere? I suggest a

better reference would be Lu et al., 2013 *Geology*: DOI:10.1130/G34488.1

Reference 5 - this reference is used to support a statement suggesting Quaternary Scandinavian ice sheet growth enhanced surface westerlies in the text (lines 59-61). However, that paper only covers the late Quaternary, focuses on short millennial scale oscillations, and makes no explicit mention of the importance of the Scandinavian ice sheet - only northern hemisphere ice mass generally.

While this is generally well written, there are quite a few ambiguous parts in the text, which often contain vaguer statements or confusing phrasing. For example:

Lines 49-54. What is the 'North Atlantic Pressure System'? Define what specifically you are talking about: NAO, westerly flow? Also, there is a question of causality here. It is implied here that summer monsoons become weak due to blocking via persistent high pressure systems, but this is ignoring the weakening of the summer monsoon due to weaker insolation and SST forcing as a driving force. Please clarify.

Lines 71-75. What is meant here by 'scenarios'? Are these scenarios to explain the development of late Oligocene/early Miocene aridity (25-22 Ma)? It's not clear from the text as the paragraph topic is on the potential drivers and not the event being referred to in the scenarios. Also, the scenarios need to be better linked to the drivers mentioned before. Are both proposed as consequences from these drivers? How would Eocene moisture have relocated the high pressure centres further east for example? Or Paratethys retreat/plateau uplift? From Fig 1c the HP centre seems to be located further south compared to Quaternary interglacial times, not so much further east. So further east in relation to what? Or is the figure wrong? When one examines Fig 1 the degree to which these scenarios are actually wildly different competing hypotheses becomes clear. This is actually not that obvious from the text - I suggest that it is clearly emphasised just how different these scenarios are and what the implications be of one over the other. This should be done to emphasise the wider significance for the broad audience of *Nature Comms*.

Line 78-79: Make it clear how this can address the problem you set up in the previous paragraphs. Does it provide the possibility to test between these two scenarios you just mentioned? These deposits have been documented before and have been analysed for grain size amongst other things. What specific implications for atmospheric circulation and aridification can already be made from the existing data and what new info can the provenance data bring to the debate. Again, this is all a bit vague and covered too cursorily.

I realise space is highly limited here but the explanations at present are confusing and rather general. Also, some of the results section contains problematic sentences with apparently missing words or confusing sentence structure. I ask that the authors could focus on trying to clarify their writing more and ensuring that there are no obvious errors such as missing words or ambiguous phrasing.

Lines 100-103: sentence has a number of errors that need fixing (the following sentence also has some errors). Please also note that these are the combined distributions from all samples in the areas mentioned. Also, it's not clear what 'statistically different' is here in relation to others. How statistically different according to the K-S statistic are the individual samples from each other (to give some sense of the sampling error) and how different are all the grouped datasets from each other? Tables of K-S statistics for comparisons of the different individual samples and source regions (not mixtures as this is shown in Fig S3) would be helpful to show some context here, and to see how much difference there is with regard to dissimilarity between all the sets of samples. Otherwise it is

hard to judge exactly how important this statistical difference is.

Fig. 3 and test lines 113-127: The term 'Plio-Quaternary deposits' used to denote areas marked yellow on the map seems highly subjective. In practice there are vast areas of Plio-Quat deposits all over China that are not marked here, notably many of the dried lake beds west of BJD and loess deposits east and south of the CLP. What marks out the ones shaded on this map? Also, the 'hypothetical' glacial dust transporting direction noted in the caption should also be marked as hypothetical in the key. I agree that it is possible given the data, but this is by no means a widely accepted view that dust transport pathways were different over the glacial versus interglacial periods - although attractive it's a debated idea. Also, why is the BJD not included in the source areas marked on the map? It appears to be included in area 1 in the text?

Regarding the choice of potential source areas and their grouping, I have a number of comments. Firstly, why is the Tarim Basin not included here in the possible source regions? I agree that much of the area is likely to have been underwater at that time but given that the conclusions of the authors suggest the presence of westerly dust transporting winds and probable aridity in Asia over this interval then perhaps it is worth considering data from this region too as many areas are likely to have been exposed, as are piedmonts of N Tibet and Tien Shan, which today feed sediment to the Tarim Basin. Many of the source areas considered here already, especially the ones north and immediately NW of the CLP are not considered to be desert/arid regions prior to the Quaternary/Pliocene (e.g., Li et al., 2013, QSR 85, 85-98 and Wang et al., 2015, P3 426, 139-158). This also makes them unlikely sources for Palaeogene dust, although I agree with the authors that it is worth examining data from these regions, given the novel conclusions in this paper. Much older however, appears to be the Taklamakan desert, potentially Oligocene in age (Zheng et al., 2015 PNAS 112, 7662-7667), but this is not included in the analysis here. The Xining mudstones are very fine grained compared to Quaternary loess, and it is feasible that the eastern Tarim and Tien Shan, N Tibet foothills could have been a source, especially considering the quite different Palaeogene topography. A good single-grain zircon U-Pb dataset exists for the Taklamakan desert so it would be easy to incorporate into the analysis (Rittner et al., 2016, EPSL 437, 127-137). Unless there is a compelling reason not to do this and to exclude this area as a potential source then I would ask that the authors analyse this data too and consider this in their interpretation. Otherwise, please explain why this area has not been included in the analysis while the northern deserts have been. The authors argue that there is a probable constancy in dust source from the Eocene to Quaternary in this submission, and given that Tarim is often argued as a probable source, especially for the Red Clay (Nie et al., 2014 EPSL 407, 35-47; Shang et al., 2016; EPSL 439, 88-100), then it should be included for that reason as well.

Secondly, the choice of groupings. Some seem very clear to me (e.g., NE Tibet), but others less so. The first group particularly is a vast area containing at least 4 separate sand sea systems, all shown to have different source signatures themselves (Bird et al., 2015; Stevens et al., 2013 - both cited in the text), and some indeed showing internal variations (e.g., Mu Us Stevens et al., 2013). Analysing all this data together would mask these differences and smooth out potentially different source signatures for the deposits in the analysis. For example, dust transport to the Xining area from the NE Mu Us (NE of Xining) would involve a fundamentally different mechanism to transport from the Badain Jaran (NW of Xining). Is it not wiser to only group source areas together for the analysis that would provide dust via the same transport pathways to the Xining area? Otherwise, the grouped

data may be mixing together more than one separate source pathway, both, neither, or either of which may have sourced the Palaeogene dust. I understand the authors have done this to try to get close to statistical representativity in their samples with as high n as possible, but this grouping should not be undertaken when there is a danger of mixing different source areas together. High n becomes meaningless if there is an unknown mixed source signal in your datasets. So, in short, how are the groupings chosen justified? Particularly group number 1? And how do you account for variation within that group in the analysis? This needs to be justified in the supplementary files extensively with recourse to specific data from the different areas.

Thirdly, while I understand that Palaeogene datasets are favoured (lines 120-123) in the analysis, this is not really comparing like with like where such data has not been used. Some of the source area groupings, especially area 1, rely on Neogene data. Isn't a fairer comparison then also with the Neogene data from NE Tibet? At least this analysis should be included to check the veracity of the comparison of the Palaeogene data from one source region, with the Neogene data from another source region, and both to the Palaeogene loess data. There is a large Neogene Yellow River dataset which integrates data from the NE Tibetan plateau (Nie et al., 2015 - cited in the text). Why not compare this Neogene data with Palaeogene data from the same region to back up the conclusions by demonstrating this difference is consistent even if Neogene deposits are only considered. There is some mention of this in the Methods but the data comparing Palaeogene versus Quaternary Northern Deserts is not shown and the explanation does not seem completely water tight: the Mu Us for example shows lots of evidence for at least part of it being derived from NE Tibetan sediments (Stevens et al., 2013), and this has also been proposed for sediments in the Alaxa sandy lands (Che and Li, 2013 Quat Res 80, 545-551), along with uplifted Gobi Altay. These areas may not have provided sediment in the same way during the Palaeogene.

While I see that a mixed local fluvial and Qaidam signature could explain the Palaeogene loess, this 1) assumes local fluvial input and 2) does not preclude other mixtures. While I see that this conclusion is later supported by the mixture modelling, at this stage in the manuscript it seems far too definite to state that 'the Qaidam basin is the only source... when mixed with local fluvial input' (lines 126-127). I ask that the authors rather present this as a hypothesis that is tested using the modelling. Also, the description of why this would be the case on the basis of age distribution data is too brief. Some clearer recourse to how the data specifically support this is needed, and some clearer consideration of alternative hypotheses. For example, the CLP data seems nearly identical to the Palaeogene loess data (Fig 2), and this is pointed out later on, yet it is clearly impossible that the proposed proximal Palaeogene fluvial sediments plus Qaidam sediments causes that pattern in the Quaternary samples. So why is a NE Tibetan integrated source not considered as a possibility for the Palaeogene, as for the Quaternary loess? At least before its further explored with mixture modelling. The way the manuscript is presented creates the impression that the conclusion is already decided before a key component of the analysis is undertaken.

With regard to the Results on lines 128-139 and mixture modelling. Table S3 seems to be labelled as Table S1 in the files (a mistake?). Fig S3 is useful but please clarify if the dissimilarity colour bar is different to the p-values mentioned in the text. Are the p-values shown anywhere? Also, I am not sure I understand the sentences lines 130-133. P values of c. 0.5 refer here to the solutions in Table S3? If the solution is nearly 3/4 local source this seems quite different to the Quaternary CLP situation where it has been argued by the authors that most of the CLP dust comes from NE Tibet

and Qaidam (Licht et al., in press). So is the sentence on lines 138-139 really correct here if a large proportion is local?

I also wonder what the best solutions (Table S3) would be if the mixture modelling was run with stepped contribution from other sources than Qaidam. Currently, fig S3 shows model runs with apparently various continuous proportions of A, B and D source regions, and steps of 10% contributions for Qaidam (C). Or is this just an artefact of the presentation? What if this was changed and region C was incorporated into one of the ternary corners and three new series of modelling runs were undertaken with either of regions A, B and D forming the 10% steps? So this would generate 4 series of Ternary diagrams each using a different 10% stepped contributions for each of the four areas (done for C right now - note that this is mislabelled as region B in the legend). I am not sure how much difference this would make (if any?) but the resulting best fit dissimilarity values for each set of runs would be useful in that they might back up or change the possible best fit outcomes suggested here. If no difference then suggest the authors note this in the supplementary info as a natural consequence of their technique.

Paragraph 141-150. I think this needs more explanation. A) What is the specific evidence that the Miocene exhumation of NE Tibet explains the difference between the Quaternary and Palaeogene dust? What ages specifically? This is vague. And b) why is it specifically former dust deposits within the Yellow River drainage area that are evoked as the source for Quat-Pliocene loess on the CLP? What is the specific evidence for this and not other deposits in the catchment; slope, lake, fluvial, alluvial fan etc, bedrock, that are highly prevalent in NE Tibet? I know the authors have published evidence for recycling of the CLP in the Quaternary, but this is not the same as recycling of dust deposits in the NE Tibetan plateau drainage area - I see no new evidence but the statements in this paragraph imply a new conclusion about the source of material entering the Yellow and Huangshui Rivers and being transported to the CLP. This needs to be explained much more and backed up with clear evidence.

Lines 174-176: agreed in part, but really you need to analyse Miocene and Pliocene dust source data to really demonstrate this. There is late Mio-Plio zircon data at least from the CLP... does it support your conclusions? This relates back to my point above about the consideration of sources that have been also evoked for Mio-Pliocene material (Tarim). If you do not analyse these data (red clay and tarim) then how can this conclusion be reached? Its strikes me that a more extensive analysis of data is needed to draw conclusions about the Miocene-Pliocene. It should not be assumed that this will be the same as the Quaternary and Eocene.

Other smaller edits:

Lines 25-27. Think this sentence is too general. 'Asian atmospheric circulation' also includes the onset of summer monsoon circulation and the theories behind this are not always the same. For example, the close of the Panama seaway (Nie et al., 2014, Sci Reports. DOI: 10.1038/srep05474). Suggest Asian arification. Or central Asian atmospheric circulation.

Lines 32 to 36. Sentence is clunky and needs rewording. Also, 'consecutive' should be 'concurrent'?

Line 41. change has to have

Line 44. Remove 'late' as you talk about interglacials and glacials in plural. Late refers to from MIS 5 to present.

Line 48. Translocated?

Line 67. Cant really use 'these' to refer to something in a previous paragraph

Line 79 - remove 'the'

Line 95 - therefore requires a

Line 97 - same approach as what? Strange phrasing.

Line 109-112 - why assume there is local input at all? Is there clear evidence that this fluvial input would have given rise to some of the aeolian deposits?

Line 141 - what does 'despite unchanged budget' mean? There are a lot of these highly condensed phrases that do not fully explain what the authors presumably intend to say.

Line 142 - Fig 3 does not show the age distributions.

157-161 - link this more clearly to the proposed scenarios in the intro.

169-171 - why not cooling?

173 - you mean Quaternary in general? Not just late Quaternary - only one glacial in the late Quat.

Line 176 - its dynamics have

In summary:

A. Summary of the key results

New evidence that modern/Quaternary atmospheric circulation in Asia was set up considerably earlier than previously thought. This implies constancy of source of dust deposits on or next to the Chinese loess plateau for 10s of millions of years and suggests that tectonic changes or land-sea organisation had little impact on synoptic conditions in Asia.

B. Originality and interest: if not novel, please give references

Wide interest and original data. I ask the authors to clarify what are the novel conclusions here in comparison to Licht et al., 2014 Nature 513, 501-506). Its not completely clear from the manuscript.

C. Data & methodology: validity of approach, quality of data, quality of presentation

Good data and methods, but some clarifications on analysis needed, and presentation could be improved.

D. Appropriate use of statistics and treatment of uncertainties

Mostly of high quality but some aspects need addressing.

E. Conclusions: robustness, validity, reliability

Generally secure but I am not convinced the source groupings are fully justified, nor the constancy of sources since the Palaeogene fully demonstrated, although I think this can be fixed with not too much effort.

F. Suggested improvements: experiments, data for possible revision

Noted fully above.

G. References: appropriate credit to previous work?

Some misplaced references but generally good.

H. Clarity and context: lucidity of abstract/summary, appropriateness of abstract, introduction and conclusions

Some overly condensed text that needs better explanation and more specific mention of data.

Suggest Methods section is reduced and moved to supplementary file to accommodate this in manuscript.

Reviewer #2 (Remarks to the Author):

This is a very timely paper that discusses the origin of the modern atmospheric configuration pattern

over eastern and central Asia. More specifically, it challenges traditional ideas favoring the onset of arid and monsoonal conditions in the late Cenozoic (associated with increased dust supply) to be linked with the uplift of the Tibetan Plateau, the retreat of the Paratethys Sea, the extension of ice sheets (or global cooling) in the northern hemisphere and/or changes in atmospheric pCO₂.

The authors give a short, but very clear explanation of the atmospheric system since the Eocene, with changing importance of high-pressure cells and westerlies. After a previous study of Eocene deposits (Licht et al. 2014), they now report on the sediment provenance of even older (Palaeogene) mudstones of partly fluvial and partly aeolian origin. Their results, based on zircon-age distributions of the mudstones and potentially contributing provenance regions, end up with the important conclusion that the Paleogene deposits show a provenance that is very similar to the one found in Quaternary loess. In addition, it is also demonstrated that aeolian dust supply dates back until the early Cenozoic (at least since 42 Ma). This dust supply may be linked with westerly winds.

Finally, the authors conclude that their results do not support previously hypothesized causes for the onset towards the modern pattern of atmospheric circulation (lines 156-168)- see above. This negative conclusion is less clearly based on data and asks for a better link with the new data.

The Methods section is clearly explaining the applied methodology.

What I miss is a description of the sample locations of the Palaeogene mudstones and sedimentological observations and interpretations of the sections that are the base for the present interpretations. It is not sufficient to refer to the resembling Eocene mudstones.

Minor remarks

-fig. 1a: It is mentioned in the text that the jet stream is located south of the Tibetan Plateau but in the figures it crosses in the central part of the Plateau.

Further, southeastern winds during spring/winter are mentioned in the text but they do not show up in the figure.

-lines: 46-48: when talking about southeastern winds it might be fair to refer to the original paper by Liu T.S. 1985, later on quantified in dust fluxes by others as, for instance, Nugteren & Vandenberghe (2004 in GI Plan Ch 41) and Vriend et al. (201 in Quat Int. 240)

-line 89: 'However': I do not see the contradiction: part of the aeolian supply is not opposing a sediment source from surrounding highlands as they may have been transported by low suspension clouds over tens of km (see Vandenberghe, 2013 in ESR 121).

Technical:

- Line 77: replace 'suggest' by 'suggests'

- Line 142: 'Fig. 3': probably you mean 'Fig. 2'?

- Line 152: Be consequent in the use of 'Myr' or 'Ma' (as in lines 29 and 94).

Jef Vandenberghe

Amsterdam 2nd March

Reviewer #3 (Remarks to the Author):

General comments:

This manuscript aims to discuss past Asian atmospheric circulations since the end of late Eocene. After read this manuscript, I have to say, there are no solid evidence for supporting their ideas. To my knowledge, many of them are speculations

Specific comments:

(1) This work is mostly based on the previous works of the first Author (Licht et al., Nature, 2014) and another work of the second-author (Dupont-Nivet et al., 2007, Nature), which they proposed that they found aeolian mudstone at Xining basin spanning 42-33 myr. This is also the basis for the discussion of this paper (please see their Lines 76-91). However, in the paper of Dupont-Nivet et al. (2007), the interbedded red mudstones (their Fig. 1c) clearly display horizontal beddings, they are not aeolian origin! But, they are fine distal fluvial or shallow-lacustrine mudstone (as the attached Figure by the reviewer)! So, it is nonsense to compare the detrital zircon ages of the paleogene "Red mudstone" at Xining to the aeolian loess on the Chinese Loess Plateau.

(2) For Figure 1, do you have any evidence or modeling results to support such paleo-atmospheric circulations? Speculations!

(3) I do not think you can only using limited detrital zircon U-Pb ages to define large-scale past wind patterns.

(4) Figure 3 is only a present site- map, how can it be used to the past of 42 myr ago?

Dr. Alexis Licht
Potsdam Universität, Germany
alicht@uni-potsdam.de

Dear editors, dear referees,

The three referees have provided constructive and judicious comments that enabled us to significantly clarify our original manuscript. We would like to particularly thank reviewer #1 whose 7 pages of detailed and thorough comments have been particularly helpful in strengthening our manuscript. Following Reviewer #1's advice, we made a significant effort to detail key points of our approach and discussion --including those related to a potential provenance from the Tarim Basin and the increased contribution of northeast Tibet during Quaternary times. These and all other points raised by the referees are specifically discussed below. With all the suggested and required additions, clarifications and corrections, a strengthened and revised manuscript is herein provided for your consideration for publication.

Reviewer #1 (Remarks to the Author):

This is an exciting paper on one of the central questions in Cenozoic climate and landscape evolution: aridity onset and set up of current circulation over Asia. The manuscript presents new single-grain provenance data from Eocene-Oligocene deposits recently identified as some of the earliest aeolian sediments in central-eastern Asia. These deposits have not universally been accepted as aeolian, despite some clear evidence presented previously, so the addition of provenance data is timely. The provenance data also serve to address two competing scenarios for Asian climatic evolution, which has implications for understanding the impact of tectonic and sea level drivers of climatic evolution. The subject matter is therefore ideal for publication in Nature Communications as it appeals to a wide range of Earth Scientists and climate researchers and underpins the climatic evolution of an important and highly populated region. If anything the authors undersell this in the submission.

For example, they do not clearly state just how fundamentally different and incompatible the Eocene-Oligocene climate scenarios outlined in the paper are. They could also make it clearer from the outset of the submission that it is not just circulation but the timing of initial aridification of the Asian landmass that is in question here.

This has been clarified and emphasized in the new manuscript (lines 68-84; see also our reply to the following comments).

I have some comments, as outlined below, and most of these are requests for clarifications or further explanations, although I also suggest some simple further modelling. Generally, the manuscript suffers a bit from being a little too 'cut down'. There is a lack of detail at key points and some of the interpretations appear to be less well substantiated in the text, even if they are once you dig a little deeper. These need to be explained more thoroughly and clearly in the text. I realise there is a strict page limit here but to me a lot more of the Methods section in the manuscript could be incorporated into the supplementary file to save manuscript space. The Methods section actually within the paper should be a very short summary. At the moment it's taking up more space that needed at the expense of more clear and reasoned analysis of the data.

We have detailed all the key points raised by reviewer #1 (see below) in the new manuscript and significantly strengthened our interpretations by additional details.

However, according to Nature Communications' guide for authors, there is no clear word limitation for the Methods section. In that sense, our Methods section does not take space at the expense of the main text (and will

be displayed at the end of the article, with a different typology). Our method section is relatively small (< 800 words) and most of the methodological aspects that are explained are important --particularly those related to the choice of the samples and the statistical treatment, as emphasized by reviewer #1 below. We thus think that this section is better in the main manuscript.

Also, the Discussion is rather disjointed at present. There is a lack of clear flow or narrative and the paragraphs not linked well together. Some conclusions seem less well supported while others are well supported but not always clearly explained. Overall, the Discussion presents lots of interesting ideas but I am not 100% sure what is specifically new in this submission, and what has not been already proposed in Licht et al., 2014 Nature 513, 501-506. The dataset is clearly new (provenance data from the Palaeogene loess deposits), but the proposal that this is loess and that it supports monsoonal and westerly flow (e.g., current/Quaternary synoptic conditions) much earlier than previously proposed were already in Licht et al., 2014. I think the dataset here is extremely important, and has big implications, however, I would like the authors to really make clear what is actually newly proposed from this data, rather than corroborating their previously proposed hypothesis. Overall I would recommend that if the authors can clarify the specific new findings here, and clarify/improve some of the writing while address some of the questions over the analysis, then this would be an exciting manuscript, and certainly publishable in Nature Communications.

The distinctions between this and our previous work are now clearly emphasized in the new manuscript (lines 85-89). Licht et al. (2014) have shown the aeolian character of Paleogene red mudstones of the Xining Basin and used this evidence for early aridification of central Asia in the middle Eocene. This early paper nonetheless did not provide any information about (1) the locus and extent of this early aridification; (2) the wind pathways associated with aeolian dust transport. Knowledge of these two key elements are critical to understand the controlling mechanisms of Asian atmospheric dynamics, because previous scenarios for the pre-Miocene atmospheric circulation (Fig. 1) proposed fundamentally different wind patterns, locus and timing for the initial aridification of the Asian interior. Tracking the provenance of the aeolian deposits of the Xining Basin allow us to provide this information and test both scenarios.

At one or two points the choice of references does not seem completely appropriate. The authors need to take care that the right references are used to support their statements. For example:
Reference 4 - this reference seems out of place in the text (line 49) as it's not directly related to summer monsoon front movement. Was this reference supposed to go elsewhere? I suggest a better reference would be Lu et al., 2013 Geology: DOI:10.1130/G34488.1
Reference 5 - this reference is used to support a statement suggesting Quaternary Scandinavian ice sheet growth enhanced surface westerlies in the text (lines 59-61). However, that paper only covers the late Quaternary, focuses on short millennial scale oscillations, and makes no explicit mention of the importance of the Scandinavian ice sheet - only northern hemisphere ice mass generally.

The references have been updated according to reviewer #1's suggestions.

While this is generally well written, there are quite a few ambiguous parts in the text, which often contain vaguer statements or confusing phrasing. For example:
Lines 49-54. What is the 'North Atlantic Pressure System'? Define what specifically you are talking about: NAO, westerly flow? Also, there is a question of causality here. It is implied here that summer monsoons become weak due to blocking via persistent high pressure systems, but this is ignoring the weakening of the summer monsoon due to weaker insolation and SST forcing as a driving force. Please clarify.

This has been clarified (line 50-56). The 'enhanced North Atlantic Pressure System' was referring to the high pressure system --enhanced after northern hemisphere cooling and ice-sheet expansion in Scandinavia and North America. We now more simply refer to 'cooling over the North Atlantic' (cf Vanderberghe et al., 2006, and

the added reference: Manabe and Broccoli, 1987).

Lines 71-75. What is meant here by 'scenarios'? Are these scenarios to explain the development of late Oligocene/early Miocene aridity (25-22 Ma)? It's not clear from the text as the paragraph topic is on the potential drivers and not the event being referred to in the scenarios. Also, the scenarios need to be better linked to the drivers mentioned before. Are both proposed as consequences from these drivers? How would Eocene moisture have relocated the high pressure centres further east for example? Or Paratethys retreat/plateau uplift? From Fig 1c the HP centre seems to be located further south compared to Quaternary interglacial times, not so much further east. So further east in relation to what? Or is the figure wrong? When one examines Fig 1 the degree to which these scenarios are actually wildly different competing hypotheses becomes clear. This is actually not that obvious from the text - I suggest that it is clearly emphasised just how different these scenarios are and what the implications be of one over the other. This should be done to emphasise the wider significance for the broad audience of Nature Comms.

This paragraph has been completely remodelled and detailed following reviewer #1's comments. Scenario #1 reflects the potential impact of changing topography and paleogeography, and is commonly evoked to explain the development of late Oligocene / early Miocene aridity (new Reference: Zheng et al. 2015, about the expansion of the Taklimakan desert). Scenario #2 reflects the impact of late Cenozoic global cooling and proposes a younger set-up for the modern central Asian atmospheric circulation (late Neogene - cf Armstrong and Allen 2011).

Line 78-79: Make it clear how this can address the problem you set up in the previous paragraphs. Does it provide the possibility to test between these two scenarios you just mentioned? These deposits have been documented before and have been analysed for grain size amongst other things. What specific implications for atmospheric circulation and aridification can already be made from the existing data and what new info can the provenance data bring to the debate. Again, this is all a bit vague and covered too cursorily. I realise space is highly limited here but the explanations at present are confusing and rather general. Also, some of the results section contains problematic sentences with apparently missing words or confusing sentence structure. I ask that the authors could focus on trying to clarify their writing more and ensuring that there are no obvious errors such as missing words or ambiguous phrasing.

We carefully revised this part now emphasizing what is new in our approach and how tracking dust provenance allows us to test both scenarios.

Lines 100-103: sentence has a number of errors that need fixing (the following sentence also has some errors). Please also note that these are the combined distributions from all samples in the areas mentioned. Also, it's not clear what 'statistically different' is here in relation to others. How statistically different according to the K-S statistic are the individual samples from each other (to give some sense of the sampling error) and how different are all the grouped datasets from each other? Tables of K-S statistics for comparisons of the different individual samples and source regions (not mixtures as this is shown in Fig S3) would be helpful to show some context here, and to see how much difference there is with regard to dissimilarity between all the sets of samples. Otherwise it is hard to judge exactly how important this statistical difference is.

We now refer to the Methods section including the full explanation. 'Statistically different' is here in the sense of the KS statistic at the 95 % confidence level. A table displaying KS statistics between all the samples is a good idea but would be extremely big and would be challenging to read. Instead, we have added a new figure (new fig. 2) displaying the Multidimensional scaling (MDS) map of all the individual samples + of the combined age distribution. MDS maps are a 2-D way to display dissimilarities (here the KS-statistic) between samples (see new reference: Vermesch 2013, as well as Stevens et al., 2013). This new figure emphasizes how fluvial clastics and red mudstones from the Xining basin are statistically different (their dissimilarity spaces do not overlap).

Fig. 3 and test lines 113-127: The term 'Plio-Quaternary deposits' used to denote areas marked yellow on the map seems highly subjective. In practice there are vast areas of Plio-Quat deposits all over China that are not marked here, notably many of the dried lake beds west of BJD and loess deposits east and south of the CLP. What marks out the ones shaded on this map? Also, the 'hypothetical' glacial dust transporting direction noted in the caption should also be marked as hypothetical in the key. I agree that it is possible given the data, but this is by no means a widely accepted view that dust transport pathways were different over the glacial versus interglacial periods - although attractive it's a debated idea. Also, why is the BJD not included in the source areas marked on the map? It appears to be included in area 1 in the text?

Fig. 3 and the text have been modified to take into account these comments.

Regarding the choice of potential source areas and their grouping, I have a number of comments. Firstly, why is the Tarim Basin not included here in the possible source regions? I agree that much of the area is likely to have been underwater at that time but given that the conclusions of the authors suggest the presence of westerly dust transporting winds and probable aridity in Asia over this interval then perhaps it is worth considering data from this region too as many areas are likely to have been exposed, as are piedmonts of N Tibet and Tien Shan, which today feed sediment to the Tarim Basin. Many of the source areas considered here already, especially the ones north and immediately NW of the CLP are not considered to be desert/arid regions prior to the Quaternary/Pliocene (e.g., Li et al., 2013, QSR 85, 85-98 and Wang et al., 2015, P3 426, 139-158). This also makes them unlikely sources for Palaeogene dust, although I agree with the authors that it is worth examining data from these regions, given the novel conclusions in this paper. Much older however, appears to be the Taklamakan desert, potentially Oligocene in age (Zheng et al., 2015 PNAS 112, 7662-7667), but this is not included in the analysis here. The Xining mudstones are very fine grained compared to Quaternary loess, and it is feasible that the eastern Tarim and Tien Shan, N Tibet foothills could have been a source, especially considering the quite different Palaeogene topography. A good single-grain zircon U-Pb dataset exists for the Taklamakan desert so it would be easy to incorporate into the analysis (Rittner et al., 2016, EPSL 437, 127-137). Unless there is a compelling reason not to do this and to exclude this area as a potential source then I would ask that the authors analyse this data too and consider this in their interpretation. Otherwise, please explain why this area has not been included in the analysis while the northern deserts have been. The authors argue that there is a probable constancy in dust source from the Eocene to Quaternary in this submission, and given that Tarim is often argued as a probable source, especially for the Red Clay (Nie et al., 2014 EPSL 407, 35-47; Shang et al., 2016; EPSL 439, 88-100), then it should be included for that reason as well.

This is an excellent point that we did not emphasize in our previous manuscript. This is now emphasized in line 130-134.

There are many different reasons that allow us to reject the Tarim area as a potential dust source during the Eocene.

-First, as mentioned by the reviewer, a significant surface of the Tarim area was episodically covered by a shallow marine - lacustrine water body during the interval of study (e.g. the work of Bosboom et al., or the Zheng et al PNAS paper). Hard to imagine how such an environment could generate dust!

-Secondly, we must here emphasize that we analysed coarse (>12 µm) aeolian zircons, and that our results are valid for the coarse fraction of the loess only (as for all the other studies using U-Pb dating to track aeolian provenance). This is now emphasized in lines 103-105. Any coarse aeolian zircon supply from the Tarim area would have to transit via saltation in low-suspension clouds across the Qaidam Basin, considering the location of the Xining Basin, at the southwesternmost margin of the CLP (map in Fig. 4). Its signature should be represented in the zircons of this region.

-Thirdly, the two papers arguing for the Tarim as a potential source for the Red Clay (Nie et al. 2014; Shang et al. 2016) are based on poorly robust age compilations (2 samples for the Tarim area, with less than 200 zircons; n < 600 for the Qaidam Basin; n < 500 for the red clays in Nie et al). Pullen et al. (2014, J. Anal. At. Spectrom. 29, 971) and Licht et al. (2016) have emphasized how age peaks are similar in central Asia and how

considerations about provenance are weak with low n ($n < 800$). We thus argue that there is, so far, no robust evidence for a past contribution of the Tarim Basin to the aeolian dust budget. This is also why we prefer not to detail the results of these two papers in our discussion.

This interestingly points out that a potential Tarim contribution in the Mio-Pliocene red clay budget should be re-investigated in the light of the new datasets of Rittner et al (2016) and Licht et al (2016), providing tons of zircon ages for the Tarim, Qaidam Basin, and the northern deserts. But this is beyond the scope of the present study.

We yet tried to investigate a potential Tarim contribution to the Eocene loess budget, following reviewer #1's concerns. There are unfortunately very few --if any-- U-Pb zircon data from Eocene samples of the area, so we used the modern database of Rittner et al (only aeolian samples from the Taklimakan: samples Tb05, 07, 11b, 20, 21, 22, 23, 27, 29, 31, 34, 39; $n_{tot} = 1256$).

Compiled Taklimakan age distribution is extremely similar to the one of the Qaidam Basin, with similar proportions for young (< 500 Ma) and for old (> 1500 Ma) age peaks (Fig R1 below); both these potential sources occupy a similar space on MDS plot (Fig R2 below). This is not surprising because (1) as proposed previously, Taklimakan coarse aeolian material might have transited via the Qaidam Basin; (2) Taklamakan desert sand dunes and Qaidam Basin fluvial clastics share the same sources, i.e. the Kunlun Mountains and neighbouring ranges in northern Tibet (e.g. Cheng et al. 2015; Rittner et al 2016).

The exact contribution from the Tarim area via the Qaidam Basin is thus tough to identify because both areas share similar sources. But:

-As exposed previously, we have good arguments that are not based on U-Pb data to reject a contribution from the Tarim Basin;

-Whatever could have been the exact contribution of the Tarim Basin, it would not change the sedimentary budget estimate provided in the results (and the consecutive interpretations) because a contribution from the Tarim area in our area of study would imply surface westerly winds blowing across the Qaidam Basin.

Fig R1: Modified Fig. 3 of the manuscript including the cumulative age distribution for the modern Taklimakan desert, dataset from Rittner et al (2016). Note the similarities with the age distribution from the Qaidam Basin.

Fig. R2: MDS map similar to new Fig. 2 in the manuscript, with modern Taklimakan aeolian data. Small icons: individual samples; big icons: compiled age distribution per source area. X and Y axis: dimensionless, see Fig. 2 for more explanation. Note that the spaces occupied by modern Tarim samples (in Yellow) and Qaidam Basin (in green) samples largely overlap.

Secondly, the choice of groupings. Some seem very clear to me (e.g., NE Tibet), but others less so. The first group particularly is a vast area containing at least 4 separate sand sea systems, all shown to have different source signatures themselves (Bird et al., 2015; Stevens et al., 2013 - both cited in the text), and some indeed showing internal variations (e.g., Mu Us Stevens et al., 2013). Analysing all this data together would mask these differences and smooth out potentially different source signatures for the deposits in the analysis. For example, dust transport to the Xining area from the NE Mu Us (NE of Xining) would involve a fundamentally different mechanism to transport from the Badain Jaran (NW of Xining). Is it not wiser to only group source areas together for the analysis that would provide dust via the same transport pathways to the Xining area? Otherwise, the grouped data may be mixing together more than one separate source pathway, both, neither, or either of which may have sourced the Palaeogene dust. I understand the authors have done this to try to get close to statistical representativity in their samples with as high n as possible, but this grouping should not be undertaken when there is a danger of mixing different source areas together. High n becomes meaningless if there is an unknown mixed source signal in your datasets. So, in short, how are the groupings chosen justified? Particularly group number 1? And how do you account for variation within that group in the analysis? This needs to be justified in the supplementary files extensively with recourse to specific data from the different areas.

This is a very good point that we have already partly emphasized in our previous publication using the same technique (Licht et al. 2016). We indeed lose sensitivity in local variations of age distribution, and the East-West changes in the Mu Us Desert are a good example of this loss of sensitivity (Stevens et al. 2013).

There is no perfect solution to select the number of potential source areas and build the 'purest' age distribution of these regions. We consider that we need at least 800 ages (Pullen et al., 2014; Licht et al. 2016) in more than 5 samples to build a regional age compilation. Best would be to have >800 ages per sample in the source areas (and not per region) and consider each sample as representative of the local U-Pb signal. However, modern datasets are far from being so complete.

Our choice of grouping samples from the Badain Jaran, Tengger and Mu Us desert appeared to be the most accurate because:

(1) considering the location of the Xining Basin (southwesternmost end of the CLP), any contribution from one of these deserts would imply a northerly component for the dust storm track;

(2) though studies have highlighted local variations in U-Pb age distribution in these deserts (e.g. Stevens et al. 2013), age compilations from the eastern deserts (BDJ and Tengger) and western deserts (Mu Us) display only slight differences (Licht et al., 2016). The main differences in detrital zircon age proportions that can be found in central China are basically between southern deserts and basins (Tarim, Qaidam, margin of northeast Tibet), that share similar sources in northeast Tibet and the Kunlun Mountains, and the northern deserts, mainly recycled from northern Asian cratons and pre-Cenozoic basement rocks.

(3) it increases the number of ages in the age compilation ($n > 1700$), thus enhancing statistical robustness.

Our choice of samples for the Qaidam Basin was limited by the U-Pb data available for the Paleogene in this area. Note that the Paleogene age compilation for the Qaidam Basin is very similar to the age compilation for the Quaternary Qaidam Basin proposed in Licht et al. (2016).

We discuss our choice of grouping for northeast Tibet in the reply to the next comment.

Finally, we would like to stress that our statistical method randomly sub-samples $n=800$ ages in each regional age compilation (for $N=200$ times). It thus takes into account a potential variability inside each age compilation, though this sub-sampling does not follow any geographical trend.

Thirdly, while I understand that Palaeogene datasets are favoured (lines 120-123) in the analysis, this is not really comparing like with like where such data has not been used. Some of the source area groupings, especially area 1, rely on Neogene data. Isn't a fairer comparison then also with the Neogene data from NE Tibet? At least this analysis should be included to check the veracity of the comparison of the Palaeogene data from one source region, with the Neogene data from another source region, and both to the Palaeogene loess data. There is a large Neogene Yellow River dataset which integrates data from the NE Tibetan plateau (Nie et al., 2015 - cited in the text). Why not compare this Neogene data with Palaeogene data from the same region to back up the conclusions by demonstrating this difference is consistent even if Neogene deposits are only considered. There is some mention of this in the Methods but the data comparing Palaeogene versus Quaternary Northern Deserts is not shown and the explanation does not seem completely water tight: the Mu Us for example shows lots of evidence for at least part of it being derived from NE Tibetan sediments (Stevens et al., 2013), and this has also been proposed for sediments in the Alaxa sandy lands (Che and Li, 2013 Quat Res 80, 545-551), along with uplifted Gobi Altay. These areas may not have provided sediment in the same way during the Palaeogene.

We chose not to integrate Neogene data from NE Tibet because dramatic changes in exhumation and incision have occurred during the Neogene in this area (Lease et al., 2007, Geology; 2012, GSA Bulletin -- quoted in the manuscript). The work of Lease et al has shown dramatic changes in U-Pb age distributions of fluvial clastics in several basins along NE Tibet --all occurring during the Neogene and related to the uplift of the Laji Shan and neighbouring ranges. In that sense, comparing our Paleogene data with Neogene fluvial clastics (from the Yellow River or other fluvial deposits) might be significantly misleading. All the samples we chose to build our age compilation for Paleogene NE Tibet correspond to periods before the identified Neogene shift in U-Pb age distributions in each basin (Lease et al., 2012).

We decided to rely on Quaternary data for the northern deserts because Paleogene clastics are mostly absent there, as explained in the Methods section. But we think that it does not impact significantly our estimates. Modern sands are mainly derived from recycling of local, pre-Cenozoic substratum as well as from minor recycling of sediment brought by Altai-sourced rivers (Chen and Li 2013; Stevens et al., 2013), areas that have likely experienced less exhumation than northeast Tibet. Indeed, though altitude of the Mongolian Plateau and Altai ranges has gradually increased since the Eocene (Caves et al., 2011, JAS), exhumation and incision appear to have been minor and localised around major faults (Glore and De Grave, 2015, Geoscience Frontiers; Jolivet et al. 2007, Geology). As highlighted by reviewer #1, part of the sediment in the western Mu Us desert displays contribution from NE Tibet due to recycling of Yellow River sediment (Stevens et al., 2013); but Alaxa sands display NE Tibet-like age distributions because they are fed by rivers flowing down the Qilian Shan, a resilient range uplifted since the early Cenozoic (e.g. Clark et al. 2010 EPSL; Duvall et al. 2011 EPSL). Despite

this minor NE Tibet contribution in the western Mu Us, we argue that Palaeogene U-Pb age distributions of the northern deserts were likely close to those of present day.

Nevertheless, we made an alternative dataset for the northern deserts only relying on U-Pb ages of pre-Cenozoic samples that today exhibit aeolian erosion features (e.g. Licht et al. 2016). But this alternative dataset just tends to exacerbate the differences between the northern deserts and the Loess by increasing the contribution of old (>1500 myr) zircons, and is less robust than the modern compilation (n=443 only). This is briefly mentioned in the Methods section.

While I see that a mixed local fluvial and Qaidam signature could explain the Palaeogene loess, this 1) assumes local fluvial input and 2) does not preclude other mixtures. While I see that this conclusion is later supported by the mixture modelling, at this stage in the manuscript it seems far too definite to state that 'the Qaidam basin is the only source... when mixed with local fluvial input' (lines 126-127). I ask that the authors rather present this as a hypothesis that is tested using the modelling.

This has been nuanced in the revised manuscript (lines 145-147).

Also, the description of why this would be the case on the basis of age distribution data is too brief. Some clearer recourse to how the data specifically support this is needed, and some clearer consideration of alternative hypotheses. For example, the CLP data seems nearly identical to the Palaeogene loess data (Fig 2), and this is pointed out later on, yet it is clearly impossible that the proposed proximal Palaeogene fluvial sediments plus Qaidam sediments causes that pattern in the Quaternary samples. So why is a NE Tibetan integrated source not considered as a possibility for the Palaeogene, as for the Quaternary loess? At least before its further explored with mixture modelling. The way the manuscript is presented creates the impression that the conclusion is already decided before a key component of the analysis is undertaken.

NE Tibet integrated source is considered as a possibility for the Palaeogene, but we reject this hypothesis because it displays a too big proportion of 1500–2500 Myr ages and a low amount of 800-1100 ages, conditions that are required for the alternative source (lines 136-138). We have emphasized these points in the new manuscript (lines 135-147), and this is later confirmed by our mixture modelling.

We discuss the similarity between CLP data and red mudstones in the discussion section (see our reply below); we now emphasize why we (and others) found a northeast Tibet signature in CLP loess, but not in the red mudstones.

With regard to the Results on lines 128-139 and mixture modelling. Table S3 seems to be labelled as Table S1 in the files (a mistake?).

This has been corrected.

Fig S3 is useful but please clarify if the dissimilarity colour bar is different to the p-values mentioned in the text. Are the p-values shown anywhere? Also, I am not sure I understand the sentences lines 130-133. P values of c. 0.5 refer here to the solutions in Table S3?

The P-value of the KS-statistic and the KS statistic itself are two different things. We use here the KS statistic (and not the p-value of the KS statistic) as a dissimilarity measure (see Vermeesch, 2013 for a detailed justification of this use). This is what is shown in the colour bar.

P-values were provided for the solutions in table S3 because they provide an alternative insight into the statistical distance between the samples. Based on the reviewer's comment, we however decided to delete all the statements regarding p-values for simplification. We now only state that the age distributions of the combinations for the best fit are statistically similar to the one of red mudstones in the sense of the KS statistic

(line 151-152). The notion of 'statistical similarity' is discussed above and present in the Methods section.

If the solution is nearly 3/4 local source this seems quite different to the Quaternary CLP situation where it has been argued by the authors that most of the CLP dust comes from NE Tibet and Qaidam (Licht et al., in press). So is the sentence on lines 138-139 really correct here if a large proportion is local?

We have clarified this statement. We argue that the sedimentary budget of the Eocene mudstones is similar to the one of Quaternary loess because both of them are made of 20-30 % of aeolian input from the Qaidam Basin and 60-70 % of recycling from fluvial deposits (lines 161-164).

The principal difference between Eocene and Quaternary budget is that the source of fluvial clastics is different. Xining drainage was apparently restricted to the surrounding ranges (Laji Shan and neighbours), whereas the Yellow River has a wide drainage basin and drains source rocks that extend deeper south in northeast Tibet (lines 201-224).

I also wonder what the best solutions (Table S3) would be if the mixture modelling was run with stepped contribution from other sources than Qaidam. Currently, fig S3 shows model runs with apparently various continuous proportions of A, B and D source regions, and steps of 10% contributions for Qaidam (C). Or is this just an artefact of the presentation? What if this was changed and region C was incorporated into one of the ternary corners and three new series of modelling runs were undertaken with either of regions A, B and D forming the 10% steps? So this would generate 4 series of Ternary diagrams each using a different 10% stepped contributions for each of the four areas (done for C right now - note that this is mislabelled as region B in the legend). I am not sure how much difference this would make (if any?) but the resulting best fit dissimilarity values for each set of runs would be useful in that they might back up or change the possible best fit outcomes suggested here. If no difference then suggest the authors note this in the supplementary info as a natural consequence of their technique.

*This is indeed an artefact of presentation. In order to simplify the presentation, we decided to display only 10 ternary diagrams for one of the regions (here C= Qaidam Basin), thus displaying only 10% stepped contributions for this particular region. **But we calculated dissimilarities to mixtures of all the possible combinations, with steps of 2% for every region (A,B,C, and D).** This is explained in more details in the paper introducing the mixture modelling strategy (Licht et al, 2016), and now detailed in the supplementary material (legend of Fig. S3).*

The combinations that best fit the red mudstones (given in Table S3) result from this process. There is no particular bias toward Qaidam Basin, we can display the same ternary diagrams with A being the region with steps of 10% contributions in the figure.

Paragraph 141-150. I think this needs more explanation. A) What is the specific evidence that the Miocene exhumation of NE Tibet explains the difference between the Quaternary and Palaeogene dust? What ages specifically? This is vague. And b) why is it specifically former dust deposits within the Yellow River drainage area that are evoked as the source for Quat-Pliocene loess on the CLP? What is the specific evidence for this and not other deposits in the catchment; slope, lake, fluvial, alluvial fan etc, bedrock, that are highly prevalent in NE Tibet? I know the authors have published evidence for recycling of the CLP in the Quaternary, but this is not the same as recycling of dust deposits in the NE Tibetan plateau drainage area - I see no new evidence but the statements in this paragraph imply a new conclusion about the source of material entering the Yellow and Huangshui Rivers and being transported to the CLP. This needs to be explained much more and backed up with clear evidence.

We have significantly strengthened this paragraph and moved it to the end of the discussion for clarification. We now explain the slight differences between the CLP and red mudstone strata, and highlight of how the Pliocene set-up of the Yellow River can potentially explain these differences by providing fluvial clastics that are

sourced deeper in northeast Tibet. The recycling of Eocene dust in the Yellow River drainage + a contribution to the Qaidam basin can additionally explain part of the CLP U-Pb signal because CLP age distribution displays an intermediate pattern between these two age distributions (Fig 2 and 3). This is more detailed in lines 206-224.

Lines 174-176: agreed in part, but really you need to analyse Miocene and Pliocene dust source data to really demonstrate this. There is late Mio-Plio zircon data at least from the CLP... does it support your conclusions? This relates back to my point above about the consideration of sources that have been also evoked for Mio-Pliocene material (Tarim). If you do not analyse these data (red clay and tarim) then how can this conclusion be reached? It strikes me that a more extensive analysis of data is needed to draw conclusions about the Miocene-Pliocene. It should not be assumed that this will be the same as the Quaternary and Eocene.

We have nuanced this statement and integrated elements of the following response in lines 166-170. At the time of submission of our manuscript, there was almost no U-Pb data from Mio-Pliocene dust (only Nie et al 2014). The recent paper of Shang et al (2016) providing much more data was not yet published. But as highlighted previously, we argue that the interpretations from these two studies should be re-interpreted in the light of the new datasets of Rittner et al (2016) and Licht et al (2016). This is yet a different (and ambitious) work to do that is out of the focus of our manuscript. We thus decided not risking over-interpretation of the Mio-Pliocene data that have been gathered so far. But note, however, that the current interpretations from these two studies --arguing for aeolian input from the Tarim Basin-- corroborate our point about the resilience of surface westerly circulation in central Asia. Paleo-wind directions in Upper Cretaceous Chinese sand dunes also corroborate this resilience.

Other smaller edits:

Lines 25-27. Think this sentence is too general. 'Asian atmospheric circulation' also includes the onset of summer monsoon circulation and the theories behind this are not always the same. For example, the close of the Panama seaway (Nie et al., 2014, Sci Reports. DOI: 10.1038/srep05474). Suggest Asian arification. Or central Asian atmospheric circulation.

Lines 32 to 36. Sentence is clunky and needs rewording. Also, 'consecutive' should be 'concurrent'?

Line 41. change has to have

Line 44. Remove 'late' as you talk about interglacials and glacials in plural. Late refers to from MIS 5 to present.

Line 48. Translocated?

Line 67. Cant really use 'these' to refer to something in a previous paragraph

Line 79 - remove 'the'

Line 95 - therefore requires a

Line 97 - same approach as what? Strange phrasing.

All these points have been taken into account.

Line 109-112 - why assume there is local input at all? Is there clear evidence that this fluvial input would have given rise to some of the aeolian deposits?

First, many studies of CLP provenance have highlighted the importance of reworking of fluvial clastics in the loess sedimentary budget (Stevens et al 2013; Nie et al 2015; Licht et al 2016). Moreover, fluvial-aeolian interactions and sediment mixing are extremely common in playa-lake sedimentary systems. See the paper of Langford (1989, Sedimentology) for an exhaustive review. There are rare, isolated wing-shaped sandstone bodies in red mudstone successions (lines 94-96 ;Zhang et al. 2015), indicating fluvial transport into the Xining playa system. Considering fluvial input as a potential dust contributor is thus natural.

Line 141 - what does 'despite unchanged budget' mean? There are a lot of these highly condensed phrases that do not fully explain what the authors presumably intend to say.

Line 142 - Fig 3 does not show the age distributions.

157-161 - link this more clearly to the proposed scenarios in the intro.

169-171 - why not cooling?

These points have been addressed and detailed accordingly.

Cooling as the main forcing mechanism to mudstone-gypsum alternations is unlikely. The 'wet phase' marked by gypsum beds has been previously interpreted as representing orbitally-controlled periods of increased rainfall (and not to a decrease of evaporation, it is actually the availability of solute and therefore water that is controlling the facies). This point has already been extensively addressed and discussed by Dupont-Nivet et al (2007) and Abels et al (2011). This increase of rainfall might be due to increased moisture supply along the existing westerly wind pathway (likely driven by warming), or a weakening or shifting of the high pressures to allow more moisture supply. Note that cooling would tend to decrease moisture transport to the area of study (with a less active hydrological cycle).

173 - you mean Quaternary in general? Not just late Quaternary - only one glacial in the late Quat.

Line 176 - its dynamics have

In summary:

A. Summary of the key results

New evidence that modern/Quaternary atmospheric circulation in Asia was set up considerably earlier than previously thought. This implies constancy of source of dust deposits on or next to the Chinese loess plateau for 10s of millions of years and suggests that tectonic changes or land-sea organisation had little impact on synoptic conditions in Asia.

B. Originality and interest: if not novel, please give references

Wide interest and original data. I ask the authors to clarify what are the novel conclusions here in comparison to Licht et al., 2014 Nature 513, 501-506). Its not completely clear from the manuscript.

C. Data & methodology: validity of approach, quality of data, quality of presentation

Good data and methods, but some clarifications on analysis needed, and presentation could be improved.

D. Appropriate use of statistics and treatment of uncertainties

Mostly of high quality but some aspects need addressing.

E. Conclusions: robustness, validity, reliability

Generally secure but I am not convinced the source groupings are fully justified, nor the constancy of sources since the Palaeogene fully demonstrated, although I think this can be fixed with not too much effort.

F. Suggested improvements: experiments, data for possible revision

Noted fully above.

G. References: appropriate credit to previous work?

Some misplaced references but generally good.

H. Clarity and context: lucidity of abstract/summary, appropriateness of abstract, introduction and conclusions

Some overly condensed text that needs better explanation and more specific mention of data. Suggest Methods section is reduced and moved to supplementary file to accommodate this in manuscript.

All these points have been carefully addressed or discussed above.

Reviewer #2 (Remarks to the Author):

This is a very timely paper that discusses the origin of the modern atmospheric configuration pattern over eastern and central Asia. More specifically, it challenges traditional ideas favoring the onset of arid and monsoonal conditions in the late Cenozoic (associated with increased dust supply) to be linked with the uplift of the Tibetan Plateau, the retreat of the Paratethys Sea, the extension of ice sheets (or global cooling) in the northern hemisphere and/or changes in atmospheric pCO₂.

The authors give a short, but very clear explanation of the atmospheric system since the Eocene, with changing importance of high-pressure cells and westerlies. After a previous study of Eocene deposits (Licht et al. 2014), they now report on the sediment provenance of even older (Palaeogene) mudstones of partly fluvial and partly aeolian origin. Their results, based on zircon-age distributions of the mudstones and potentially contributing provenance regions, end up with the important conclusion that the Paleogene deposits show a provenance that is very similar to the one found in Quaternary loess. In addition, it is also demonstrated that aeolian dust supply dates back until the early Cenozoic (at least since 42 Ma). This dust supply may be linked with westerly winds.

Finally, the authors conclude that their results do not support previously hypothesized causes for the onset towards the modern pattern of atmospheric circulation (lines 156-168)- see above. This negative conclusion is less clearly based on data and asks for a better link with the new data.

As previously exposed, we have clarified and strengthened these paragraphs. The proposed scenarios and their link to regional atmospheric circulation are now more clearly explained (lines 68-84). Our interpretations are now directly linked to the proposed scenarios (lines 169-175).

The Methods section is clearly explaining the applied methodology.

What I miss is a description of the sample locations of the Palaeogene mudstones and sedimentological observations and interpretations of the sections that are the base for the present interpretations. It is not sufficient to refer to the resembling Eocene mudstones.

This is a good point that we have indeed forgotten to mention in the main manuscript: the sedimentological context of the samples was previously described in details in various publications (Dupont-nivet et al., 2007; Abels et al., 2011; Bosboom et al. 2014; Cheng et al 2015). The supplementary material provides lithology, GPS coordinates and ages for all the samples used in the study, as well as key references describing the sections. This is now mentioned in the main text (lines 234-237).

Minor remarks

-fig. 1a: It is mentioned in the text that the jet stream is located south of the Tibetan Plateau but in the figures it crosses in the central part of the Plateau.

The Fig. 1a displays the annually average position of the jet stream. As the jet stream seasonally moves northwards (in Summer) and southwards (in Winter) of the Tibetan Plateau, its average location crosses the central part of the Plateau.

Further, southeastern winds during spring/winter are mentioned in the text but they do not show up in the figure.
-lines: 46-48: when talking about southeastern winds it might be fair to refer to the original paper by Liu T.S. 1985, later on quantified in dust fluxes by others as, for instance, Nugteren & Vandenberghe (2004 in GI Plan Ch 41) and Vriend et al. (201 in Quat Int. 240)

-line 89: 'However': I do not see the contradiction: part of the aeolian supply is not opposing a sediment source from surrounding highlands as they may have been transported by low suspension clouds over tens of km (see Vandenberghe, 2013 in ESR 121).

Technical:

- Line 77: replace 'suggest' by 'suggests'

- Line 142: 'Fig. 3': probably you mean 'Fig. 2'?

- Line 152: Be consequent in the use of 'Myr' or 'Ma' (as in lines 29 and 94).

Jef Vandenberghe

Amsterdam 2nd March

All these changes were taken into account. We have tried to limit the number of references, and the paper of Roe

(2009) is a good synthesis of modern spring/winter atmospheric circulation in central China.

Reviewer #3 (Remarks to the Author):

General comments:

This manuscript aims to discuss past Asian atmospheric circulations since the end of late Eocene. After read this manuscript, I have to say, there are no solid evidence for supporting their ideas. To my knowledge, many of them are speculations

Specific comments:

(1) This work is mostly based on the previous works of the first Author (Licht et al., Nature, 2014) and another work of the second-author (Dupont-Nivet et al., 2007, Nature), which they proposed that they found aeolian mudstone at Xining basin spanning 42-33 myr. This is also the basis for the discussion of this paper (please see their Lines 76-91). However, in the paper of Dupont-Nivet et al. (2007), the interbedded red mudstones (their Fig. 1c) clearly display horizontal beddings, they are not aeolian origin! But, they are fine distal fluvial or shallow-lacustrine mudstone (as the attached Figure by the reviewer)! So, it is nonsense to compare the detrital zircon ages of the paleogene "Red mudstone" at Xining to the aeolian loess on the Chinese Loess Plateau.

There are indeed horizontal beddings in the sections, most of them being gypsum beds. This has been described in details in Abels et al (2011). There is also sparse evidence of beddings in the red mudstones (though very rare, red mudstones are almost exclusively massive).

However, evidence of beddings or sedimentary figures in the red mudstones is not contradictory with their aeolian nature. In playa-lake sedimentary systems, reworking of aeolian dust and mixing with fluvial clastics is common (see our previous comments above). This is a common feature in modern playa-systems (Langford, 1989, Sedimentology) and has been described in past aeolian dust deposits, including in the red clays of central China (Alonso-Zarza et al. 2009 Sedimentary Geology and the heated discussion that followed in Guo et al., 2010 and in Oldfield and Bloemendal, 2011).

Aeolian dust is commonly transported into environments with soils, lakes and distal alluvial fans and is incorporated in the local deposits. This is exactly what is argued for the Xining Basin, a Playa lake / saline lake environment with aeolian dust input (Abels et al. 2011; Licht et al. 2014). This interpretation is in line with the results of this manuscript, that highlight mixing between fluvial input and aeolian input in the red mudstones.

(2) For Figure 1, do you have any evidence or modeling results to support such paleo-atmospheric circulations? Speculations!

All the different subfigures of figure 1 are synthesized from previous geological studies and climate simulations for the glacial and interglacial patterns (Roe, 2009; Lu et al., 2013, Broccoli and Manabe, 1987; Vandenberghe et al., 2006; Jiang and Lang, 2010; Kapp et al. 2011; Pullen et al. 2011) as well as for the pre-Miocene scenarios (Ramstein et al. 1997; Zhang et al. 2012; Armstrong and Allen, 2011; Zheng et al. 2015). All these references are provided in the manuscript. There are, of course, many other papers supporting these four syntheses and we will be glad to provide more references to reviewer #3 if she/he is not convinced by all these papers.

(3) I do not think you can only using limited detrital zircon U-Pb ages to define large-scale past wind patterns.

Determining dust provenance via U-Pb of aeolian dust zircons allows to reconstruct dust storm pathways. These dust storm pathways are linked to regional surface wind patterns. This approach is common in loess provenance

studies (e.g. Pullen et al. 2011; Stevens et al. 2013; Bird et al. 2015; Licht et al. 2016) and is now better explained in lines 101-105.. All these references are given in the main manuscript and are just few examples of what has been recently done with this technique.

Regarding the 'limited' character of our dataset: actually, our dataset provides more ages than most of the previous studies using the same approach (cf the discussion above about the papers of Nie et al 2014 and Shang et al 2016 on the red clays). We provide $n > 1200$ ages for the red mudstones, an amount of ages that can be considered as particularly robust compared to other available datasets (eg. the discussions in Pullen et al. 2014 and Licht et al 2016).

(4) Figure 3 is only a present site- map, how can it be used to the past of 42 myr ago?

Fig. 3 (now Fig. 2) is provided to show the location of our samples relative to the modern Xining Basin. There has been several major paleogeographic changes in central Asia during the last 42 Myr (retreat of the Paratethys sea, uplift of northeastern Tibet --all of them being schematically shown on Fig. 1 and explained in the main text), but no major regional rotation or block extrusion, as seen in southeast Asia for instance. Even the most extreme paleogeographic reconstructions of Cenozoic Asia in terms of extrusion and rotation (Replumaz and Tapponnier 2003, JGR, for instance) do not modify the location of the Qaidam Basin and northeast Tibet relative to the northern China cratons. In that sense, the location of samples relative to the Xining Basin in the figure can be considered as valid for the Eocene. Illustrated paleogeographic reconstructions of central Asia during the Eocene have been proposed in Lippert et al (2014, GSA special paper), Licht et al (2014) and Bosboom et al (2015).

REVIEWERS' COMMENTS:

Reviewer #1 (Remarks to the Author):

I thank the authors for their detailed comments and consideration of my suggestions from the first review. These have clarified many of my questions and greatly improved the manuscript. I am generally happy with the authors' responses regarding grouping of the source data and the Tarim basin, although I think that it is still open to question as to the degree to which it is better to go for high n rather than focus on lower n but not group separate possible sources. As the authors state, the ideal scenario is that each source area has high n but this state has not yet been reached and I generally agree with the authors approach to get around this as best possible in the meantime. The significance of the manuscript is now expressed much more clearly, and the difference from previous work is emphasized. In fact, the conclusions are extremely striking and very exciting. So I recommend publication in Nature Communications. Some minor comments to address prior to final publication are presented below.

34: 'stability of synoptic-level...'

81: 'would then have shifted the zone of peak sea surface temperatures southward'

83: 'patterns as well as the geographic position and...'

85: 'Eocene aeolian dust...'

105 and elsewhere: "saltation in suspension clouds" seems an oxymoron. Saltation is restricted to a few 10s of cms from the sediment surface. So the processes operating with silt transport >12 microns is rather low level (lower tropospheric) suspension after larger saltating grains (sand sized) impact the sediment surface and repeatedly re-entrain silt sediment. See Ujvari et al., 2016 ESR 154, 247-278. The use of saltation to describe transport of particles >12 microns is a bit misleading in general (line 133 e.g.). I agree with the principle here that transport over such long distances for relatively coarse silt particles is unlikely (but it happens - see research of Jan-Berend Stuut on presence of coarse silt particles from African found in marine cores and sediment traps in the far west Atlantic), but nonetheless most of the zircons studied here are likely to have undergone low level suspension rather than saltation. This should be only for a few km to 100s of km but this is fundamentally different process to saltation - which is bouncing along the sediment surface for 10s of metres at a few m max height. These saltating grains are sands that impact the surface and emit silt and clay particles... which then get suspended for longer periods. I guess the authors mean that the silt particles would need to be repeatedly entrained over short term, lower atmospheric suspension across the Qaidam basin, presumably via saltating impactor grains during dust/sand storms.

165: for clarity I suggest adding 'albeit via different fluvial systems' or similar.

199: 'and atmosphere'

206: 'implications for'

207: clarify what you mean by 'unchanged budget' here. Also add 'zircon age' before 'distributions'

249: 'age compilations'

259: 'does not' and 'age distributions'

Thomas Stevens

Reviewer #2 (Remarks to the Author):

I have looked again at the manuscript and -as I was already positive after the first version- I looked especially at the revised parts following the comments by reviewers.

The first reviewer asked especially for clarifications and details at particular points. The authors have made a serious effort to cope with all concerned remarks. Therefore, the paper is now stronger and clearer than it was before. The changes and additions are to the point and thus improve the paper even more. Especially the potential provenance regions (Tarim, Qaidam) are discussed in greater detail now. I am not so sure the extension at the end of the paper (l. 206-224) is really necessary. Also I had a few requests for explanation or more precision. I am satisfied with the amendments although I am not convinced of the 'synthetic value' of the paper by Roe on the provenance of aeolian sediments. But I do not insist on the latter point.

I was surprised to read the comments of reviewer 3 on the first version of the manuscript. I feel that the authors have given a very solid evidence to support their ideas. As to the first specific comment of the reviewer3, I fully agree with the reply of the authors that there is no incompatibility between a primary aeolian origin of the sediments and their posterior (secondary) reworking in lakes or by rivers leading to other sedimentary structures than original homogeneous structures of wind deposits. In that sense, the discussions by Alonso-Zarza and Guo are rather semantic than fundamentally different. I am also satisfied with the answers to the comments 2-4 by the authors.

Again, in my opinion, the evidence provided for largely unchanged surface western paleo-winds and accompanying circulation systems since Eocene times is very convincing and is highlighted extensively in final part of the Discussion (l. 191-205). They oppose traditional previous hypotheses and thus open new scientific perspectives.

Dr. Alexis Licht
Potsdam Universität, Germany
alicht@uni-potsdam.de

Response to referees' comments on the revised manuscript

Reviewer #1 (Remarks to the Author):

I thank the authors for their detailed comments and consideration of my suggestions from the first review. These have clarified many of my questions and greatly improved the manuscript. I am generally happy with the authors' responses regarding grouping of the source data and the Tarim basin, although I think that it is still open to question as to the degree to which it is better to go for high n rather than focus on lower n but not group separate possible sources. As the authors state, the ideal scenario is that each source area has high n but this state has not yet been reached and I generally agree with the authors approach to get around this as best possible in the meantime. The significance of the manuscript is now expressed much more clearly, and the difference from previous work is emphasized. In fact, the conclusions are extremely striking and very exciting. So I recommend publication in Nature Communications. Some minor comments to address prior to final publication are presented below.

34: 'stability of synoptic-level...'

81: 'would then have shifted the zone of peak sea surface temperatures southward'

83: 'patterns as well as the geographic position and...'

85: 'Eocene aeolian dust...'

105 and elsewhere: "saltation in suspension clouds" seems an oxymoron. Saltation is restricted to a few 10s of cms from the sediment surface. So the processes operating with silt transport >12 microns is rather low level (lower tropospheric) suspension after larger saltating grains (sand sized) impact the sediment surface and repeatedly re-entrain silt sediment. See Ujvari et al., 2016 ESR 154, 247-278. The use of saltation to describe transport of particles >12 microns is a bit misleading in general (line 133 e.g.). I agree with the principle here that transport over such long distances for relatively coarse silt particles is unlikely (but it happens - see research of Jan-Berend Stuut on presence of coarse silt particles from African found in marine cores and sediment traps in the far west Atlantic), but nonetheless most of the zircons studied here are likely to have undergone low level suspension rather than saltation. This should be only for a few km to 100s of km but this is fundamentally different process to saltation - which is bouncing along the sediment surface for 10s of metres at a few m max height. These saltating grains are sands that impact the surface and emit silt and clay particles... which then get suspended for longer periods. I guess the authors mean that the silt particles would need to be repeatedly entrained over short term, lower atmospheric suspension across the Qaidam basin, presumably via saltating impactor grains during dust/sand storms.

165: for clarity I suggest adding 'albeit via different fluvial systems' or similar.

199: 'and atmosphere'

206: 'implications for'

207: clarify what you mean by 'unchanged budget' here. Also add 'zircon age' before 'distributions'

249: 'age compilations'

259: 'does not' and 'age distributions'

All these edits have been taken into account in the revised manuscript. We clarified our statement about low level suspension transport of particles following the referee's comment.

Reviewer #2 (Remarks to the Author):

I have looked again at the manuscript and -as I was already positive after the first version- I looked especially at

the revised parts following the comments by reviewers.

The first reviewer asked especially for clarifications and details at particular points. The authors have made a serious effort to cope with all concerned remarks. Therefore, the paper is now stronger and clearer than it was before. The changes and additions are to the point and thus improve the paper even more. Especially the potential provenance regions (Tarim, Qaidam) are discussed in greater detail now. I am not so sure the extension at the end of the paper (l. 206-224) is really necessary.

Also I had a few requests for explanation or more precision. I am satisfied with the amendments although I am not convinced of the 'synthetic value' of the paper by Roe on the provenance of aeolian sediments. But I do not insist on the latter point.

I was surprised to read the comments of reviewer 3 on the first version of the manuscript. I feel that the authors have given a very solid evidence to support their ideas. As to the first specific comment of the reviewer3, I fully agree with the reply of the authors that there is no incompatibility between a primary aeolian origin of the sediments and their posterior (secondary) reworking in lakes or by rivers leading to other sedimentary structures than original homogeneous structures of wind deposits. In that sense, the discussions by Alonso-Zarza and Guo are rather semantic than fundamentally different. I am also satisfied with the answers to the comments 2-4 by the authors.

Again, in my opinion, the evidence provided for largely unchanged surface western paleo-winds and accompanying circulation systems since Eocene times is very convincing and is highlighted extensively in final part of the Discussion (l. 191-205). They oppose traditional previous hypotheses and thus open new scientific perspectives.

We thank reviewer #2 for his enthusiasm. We think that the extension at the end of the paper (on the Quaternary implications of our study) is important because 1) it proposes a new interpretation for the mechanisms of recent dust supply into central China; 2) most of all loess studies are led by Quaternary geologists, and they will be particularly interested by this topic.